# Plant-Derived Natural Products as Lead Agents against Common Respiratory Diseases

**DOI:** 10.3390/molecules27103054

**Published:** 2022-05-10

**Authors:** Ayodeji Oluwabunmi Oriola, Adebola Omowunmi Oyedeji

**Affiliations:** Department of Chemical and Physical Sciences, Faculty of Natural Sciences, Walter Sisulu University, Nelson Mandela Drive, P/Bag X1, Mthatha 5117, South Africa; aoyedeji@wsu.ac.za

**Keywords:** respiratory diseases, natural products, plant-derived compounds, lead molecules

## Abstract

Never has the world been more challenged by respiratory diseases (RDs) than it has witnessed in the last few decades. This is evident in the plethora of acute and chronic respiratory conditions, ranging from asthma and chronic obstructive pulmonary disease (COPD) to multidrug-resistant tuberculosis, pneumonia, influenza, and more recently, the novel coronavirus (COVID-19) disease. Unfortunately, the emergence of drug-resistant strains of pathogens, drug toxicity and side effects are drawbacks to effective chemotherapeutic management of RDs; hence, our focus on natural sources because of their unique chemical diversities and novel therapeutic applications. This review provides a summary on some common RDs, their management strategies, and the prospect of plant-derived natural products in the search for new drugs against common respiratory diseases.

## 1. Introduction

The human respiratory system, otherwise known as the ventilatory system, is a biological system made up of specific organs and structures such as oropharyngeal and nasopharyngeal cavities, larynx, and trachea (upper respiratory tract), and the lower respiratory tract, which includes bronchi, lungs, and diaphragm [1,2,3]. The lungs are an important part of the respiratory system that facilitate gas exchange from the environment into the bloodstream for healthy living [4].

The respiratory functions can be hampered by infections and diseases of the lungs and their associated organs and structures. These diseases are categorised as obstructive and restrictive lung diseases [4]. Examples of obstructive lung diseases are asthma and chronic obstructive pulmonary disorder (chronic bronchitis), while restrictive lung diseases include idiopathic pulmonary fibrosis, pneumoconiosis, and sarcoidosis [5] (Figure 1). These respiratory diseases (RDs), otherwise referred to as “respiratory disorders”, “airways diseases”, “pulmonary diseases” or lung diseases, often arise from bacterial and viral infections in the upper and lower respiratory tracts, causing the common cold, otitis, sinusitis, pharyngitis, epiglottitis, laryngotracheitis, bronchitis, bronchiolitis, and pneumonia [6]. Some of the causative agents include *Mycobacterium tuberculosis*, *Haemophilus influenza* type b., *Streptococcus pyogenes*, *Chlamydia* sp., and *Candida albicans* [7] (Table 1).

Some viral infections of the respiratory tracts are implicated in disease pandemics such as influenza (flu), Middle east respiratory syndrome coronavirus (MERS-CoV), and more recently, the novel severe acute respiratory syndrome coronavirus 2 (SARS-CoV-2) infection-causing COVID-19 disease that is currently ravaging the world [8]. SARS-CoV and MERS-CoV broke out in 2003 and 2012, respectively [9], and influenza claimed about 389,000 lives globally in the year 2017 [10]. As of 22 January 2022, about 5.59 million people had died from 346 million reported cases of COVID-19 globally [11]. These show the significant negative impacts of RDs on individual lives, national development, and human existence.

Inhalation therapy is one of the oldest approaches, yet still relevant to the management of RDs, dating back to more than 2000 years of Ayurvedic medicine in India [12]. It is centred around delivery of high pulmonary drug concentrations to the lungs, at low inhaled doses; thus, it offers substantial efficacy while simultaneously reducing the risk of side effects associated with many orally or intravenously administered drug doses [13,14]. Recent advances in the management of RDs include vaccination; use of drugs such as antibiotics, agonists, and cortisones; ventilatory support; inhalation therapy, and lung surgery [15]. Unfortunately, new RDs continue to emerge, while multi-drug resistant strains and variants of pathogens continue to render many available drugs ineffective [16,17], hence the need for more efficacious and less toxic anti-infective agents that could help to reduce the burden of RDs and forestall seasonal or regular disease pandemics in the nearest future.

The role of natural products (NPs) in drug discovery cannot be over-emphasized. NPs are chemical substances produced by living organisms such as plants, animals, and marine organisms [18]. They are primary and secondary metabolites and may only be isolatable in small quantities from natural sources [19]. Structurally, they range from small molecules, such as thymol, thymoquinone and penicillin, to complex molecules such as tachyplesin I and II, with unique chemical and biological properties (Figure 2). They are important leads to new drugs and are thus categorised as drug candidates [20,21,22].

NPs are regarded as the hallmark of modern pharmaceutical care because they continue to provide new leads with novel biological mechanisms of action against emerging diseases [23]. Currently, about 60% of drugs in the market worldwide are natural product-derived [24], which underlines the significance of NPs in the discovery of new drugs.

This review puts respiratory diseases (RDs) in perspective and summarises the prospect of plant-derived natural products in the discovery of new drugs against some common respiratory diseases, including asthma, COPD, tuberculosis, pneumonia, influenza, and COVID-19.

**Figure 1 molecules-27-03054-f001:**
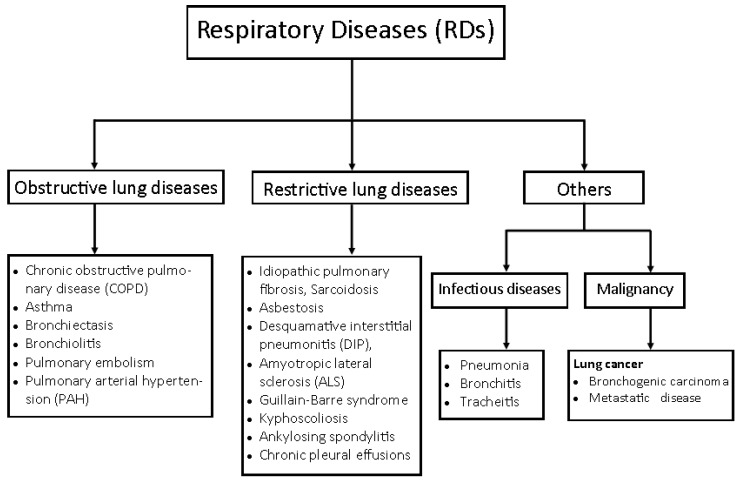
Group of respiratory diseases (RDs) adapted with permission from Kritek and Choi [25]. Copyright 2016, Basicmedical key.

**Figure 2 molecules-27-03054-f002:**
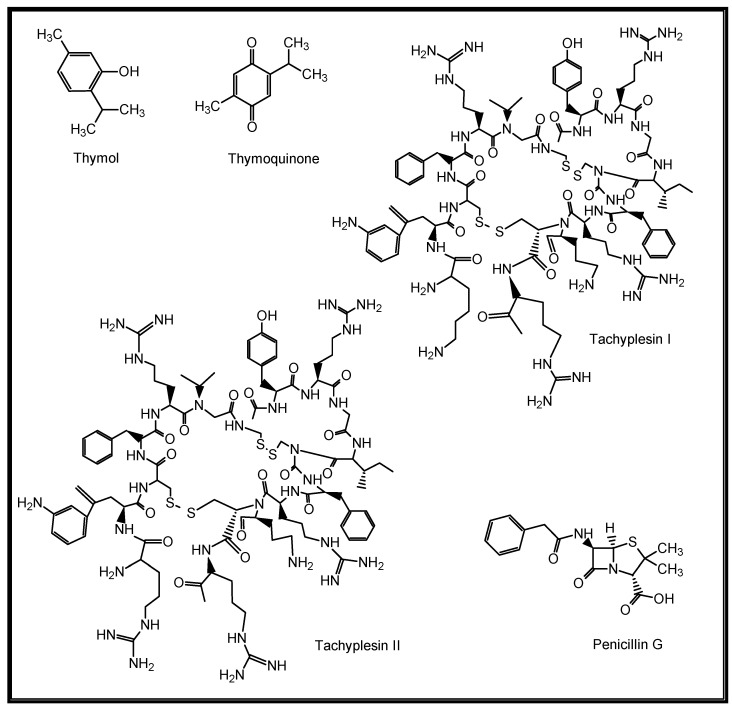
Structures of some naturally occurring bioactive compounds.

## 2. Methodology

An extensive literature survey of published articles was conducted, using scientific databases such as Google Scholar, Mendeley, PubMed, ScienceDirect and Scopus. The keywords included: “respiratory diseases”, “common respiratory diseases” and “natural products” searched singly and in pairs; “medicinal plant agents for respiratory diseases”; “natural compounds for common respiratory diseases”; and “prospects of natural products in the management of common respiratory diseases”. Publications were obtained from the literature search until March, 2022 with no language restriction. Data inclusion criteria included: reports on natural or nature-inspired compounds implicated against respiratory diseases (RDs); and reports indicating lung, airways and pulmonary disorders or diseases and their diagnostic and management strategies. Data exclusion criteria were reports on the same natural compounds with the same biological activities linked to respiratory diseases, to avoid duplication, and reports on natural compounds against diseases other than RDs as well as those biological activities (e.g., antidiabetic, antimalaria, neuroprotective and insecticidal) not directly linked to respiratory diseases. The structures of compounds were generated using the ChemDraw Ultra^®^ 7.0 software package, CambridgeSoft Corporation (Cambridge, MA 02140, USA).

This review may be the first to highlight the major categories of respiratory diseases, their therapeutic management, and the prospect of natural products of plant origin in the discovery of new drug candidates against some common RDs such as asthma, COPD, TB, pneumonia, influenza, and COVID-19.

## 3. Respiratory Diseases in Focus

### 3.1. Risk Factors and Epidemiology

Respiratory diseases (RDs) are illnesses that affect the organs and tissues in the lungs and airways, limiting gas exchange and breathing [26]. They include acute respiratory infections and chronic respiratory diseases (CRDs) such as influenza, pneumonia, bronchitis, lung cancer, and chronic obstructive pulmonary disorder (COPD) [27]. CRDs are diseases of the airways and other structures of the lungs, such as asthma, chronic obstructive pulmonary disease (COPD), pulmonary fibrosis, occupational lung diseases, interstitial lung disease, and pulmonary hypertension amongst others [28] (Figure 1).

Common risk factors for RDs include tobacco smoking, high levels of air pollution, occupational hazards (dusts and chemicals), urbanization, industrialization, poor socioeconomic and health services, the HIV/AIDS epidemic, and respiratory tract infections and genetics [29,30,31] (Figure 3). Most upper respiratory tract infections are of viral aetiology except for a few caused by pathogenic bacteria, such as epiglottitis and laryngotracheitis caused by *Haemophilus influenza* type b, and pharyngitis often caused by *Streptococcus pyogenes* [32]. For example, the common cold is a viral respiratory infection caused by rhinoviruses, coronavirus, parainfluenza viruses, influenza virus, adenoviruses, and respiratory syncytial virus [32]. Some common respiratory diseases, their global morbidity ranking (disease burden), and causative agents are presented in Table 1, while Figure 3 shows the major factors responsible for RDs.

Respiratory diseases (RDs) constitute a major public health problem worldwide. Ferkol and Schraufnagel highlighted five conditions that primarily contribute to the global burden of respiratory diseases, which are asthma, COPD, acute respiratory infections, tuberculosis, and lung cancer [47].

In the last one decade, an average of 65 million people suffered from moderate to severe levels of COPD, from which 3 million people die each year [48]. About 334 million people suffer from asthma, and influenza virus-related diseases are responsible for up to 400,000 deaths annually; tuberculosis caused about 1.4 million deaths in 2015, and about 1.6 million people die annually from lung cancer. More than 100 million people suffer from sleep-disordered breathing, and over 50 million people struggle with occupational lung diseases, while millions of people live with pulmonary hypertension [48]. Bronchial asthma and COPD account for a significant burden in low- and middle-income countries [28].

Furthermore, in 2017, about 540 million people in the world suffered from chronic respiratory diseases, such as asthma, COPD, pulmonary sarcoidosis, interstitial lung disease, silicosis, asbestosis, which is an increase of about 40% when compared to the 1990 figures [49,50]. Mortality due to these diseases stood at 3.9 million in 2017, which is about 18% increase since the 1990 report [34]. Tobacco smoking accounted for the most prevalent cause of disability due to RDs in men worldwide, while the leading risk factors for disability in women were household air pollution from solid fuel use in South Asia and sub-Saharan Africa, exposure to ambient particulate matter in Southeast Asia, East Asia, Oceania, and the developed regions within North Africa and Middle East, and smoking in all other developed regions [36].

In sub-Saharan Africa, about 83% of residents still use solid fuel for cooking, which, according to the WHO, is responsible for 130/100,000 deaths in the region [50]. Many countries are still battling rising epidemics of tobacco smoking among women and adolescents, while the levels of ambient particulate matter pollution, ambient ozone pollution, and several occupational exposures increased significantly between 1990 and 2017 [51,52]. A report has also shown the significant association between ambient temperature and the burden of morbidity caused by RDs within the sub-tropics, where it was found that exposure to non-optimal temperatures increased the risk of respiratory morbidity, and moderate heat contributes significantly to the morbidities of temperature related RDs [53].

Respiratory diseases are known to affect significantly the infant and young population across the globe. For instance, pneumonia has been reported to cause about 1.3 million childhood deaths every year, asthma is the commonest non-communicable disease in children, while paediatric tuberculosis (TB) constitutes up to 20% of the TB morbidity in high-incidence countries [54]. There are up to 10 million new cases of clinical TB globally and 1.5 million deaths every year, with the emergence of drug-resistant TB considered as a major public health crisis [55].

It is noteworthy that respiratory tract infections contribute significantly to morbidity and mortality of RDs. According to the WHO, lower respiratory infection is the most common infectious disease-causing deaths worldwide, accounting for the loss of about 3.46 million deaths annually, with most of them being children in developing countries [56].

Respiratory diseases impose a huge economic burden on both industrialized and developing countries [57]. RDs cost the United Kingdom about GBP 11.1 billion in 2014, which comprised GBP 9.9 billion in treatment costs and an estimated GBP 1.2 billion loss in productivity [58]. Annual expenditures for workers in the United States summed up to USD 7 billion for asthma care and USD 5 billion for COPD care between 2011 and 2015, according to the Centre for Disease Control and Prevention (CDC) report [59]. The annual cost of treatment for patients with an RD in the Asia-Pacific region was estimated at USD 4191 per patient, while the mean annual cost of treatment for patients who reported lung impairment at work was USD 7315 in 2014, thus resulting in a 36% reduction in productivity [60]. In the Central Asian countries (Eurasia) such as Kyrgyzstan, Uzbekistan, Tajikistan, Kazakhstan and Turkmenistan, the prevalence of TB is still significantly high, placing enormous costs on the government and patients [61]. In sub-Saharan Africa, deaths associated with solid fuel rose by 18% between 1990 and 2013 and cost the African economy approximately USD 232 billion by the year 2013 [62].

### 3.2. Management Strategies

The effective treatment of RDs will reduce the disease burden and health care costs and improve quality of life and productivity [60]. Over the last few decades, RDs have been managed by vaccination, hormone therapy such as the use of corticosteroids, targeted therapy such as the use of nebulizer, and chemotherapy such as the use of antibiotics (antibacterial and antifungal drugs) and antiviral and cytotoxic drugs [63]. However, many of these management regimens have their complications and adverse side effects. The continuous emergence of novel respiratory infections such as SAR-CoV-2, the resurgence of new strains of pathogens such as the multi-drug resistant strains of TB, and the complexities in their mode of action, are major setbacks to effective vaccination [64,65]. Hormone therapy has been reported to be associated with increased risk of asthma among pre-menopausal women [66].

In recent times, many of the currently approved drugs have presented with numerous complications. For instance, bleomycin has been reported to cause inflammation of the lungs (lung fibrosis) and breathlessness, among the most common clinical presentations of which are interstitial pneumonitis/fibrosis, hypersensitivity pneumonitis, and capillary leak syndrome [67]. Beta (2)-agonists administered as therapy for asthma and COPD have recognised systemic sequelae, such as hypokalaemia and chronotropic effects that may be life-threatening in susceptible patients [68]. Intravenous RSV immunoglobulin has an unfavourable risk–benefit balance, particularly with the availability of monoclonal antibodies, while the use of corticosteroids is limited nowadays because of reported cases of adrenal suppression, especially in children [69,70]. Inhaled corticosteroids, which are commonly used in combination with long-acting β2-agonists to reduce exacerbation rate in worsening cases of COPD, have been associated with increased risk of pneumonia [71]. There has been an increasing rate of macrolide-resistant bacteria in the last few decades, making the use of drugs such as macrolides in combination with β-lactam agent ineffective for the treatment of community-acquired pneumonia [72]. Anti-TB drugs such as ethambutol, isoniazid, pyrazinamide, rifampin, and streptomycin have common side effects, which include cutaneous reactions, gastrointestinal intolerance, haematological reactions, and kidney failure [73]. Some FDA-approved anti-influenza drugs such as baloxavir marboxil, peramivir, oseltamivir phosphate and zanamivir, have proven ineffective in patients with serious influenza requiring hospitalization, and with common side effects such as bronchitis, common cold, diarrhoea, headache, and nausea [74]. Dexamethasone is a corticosteroid, anti-inflammatory and immunosuppressive drug that has been widely used in recent years for the management of COVID-19 patients who are in critical condition. However, the common side effects of this drug include acne, depression, dizziness, gastritis, headache, insomnia, restlessness, and vomiting [75]. Azithromycin is an antibiotic drug that received much attention for clinical use among patients with early-stage SARS-CoV-2 infection. However, some adverse reactions such as abdominal pain, anaphylaxis, diarrhoea, nausea, QT prolongation, *Clostridium difficile* infection and vomiting have been reported with azithromycin [75]. Thus, there is a need to exploit nature for its bioactive agents, which can be optimized for the chemotherapeutic treatment of the afore-mentioned RDs and many more with little or no side effect.

## 4. Plant-Derived Natural Products as Lead Agents against Common Respiratory Diseases

Natural products (NPs) are generally described as chemical substances produced by living organisms that are found in nature [18]. Natural product sources include plants, animals, lower and marine organisms, and minerals [76]. Plant-derived NPs are characterised by enormous chemical entities with structural complexities, which are optimized by evolution to serve specific biological functions [77]. These structurally diverse substances, also referred to as metabolites, especially secondary metabolites (terpenoids, alkaloids, flavonoids, coumarins, anthraquinones, saponins, phenolics and phenolic glycosides), produce physiological actions in man; thus, offering great therapeutic value [78].

Historically, they continue to play a key role in the therapeutic armoury of mankind against diseases. The earliest records of NPs were plant extracts such as oils from Cypress (*Cupressus sempervirens*) and *Myrrh* (*Commiphora* species), inscribed on clay tablets in cuneiforms from Mesopotamia (2600 B.C.), which are still used today ethnomedicinally to treat colds, coughs, and inflammation [78]. The Egyptian pharmaceutical record (Ebers Papyrus, 2900 B.C.) documented over 700 plant-based drugs in the forms of gargles, infusions, ointments and pills [79]. The Chinese Materia Medica (1100 B.C.) contained 52 prescriptions, the Shennong Herbal (~100 B.C.) enlisted 365 drugs, while the Tang Herbal (659 A.D.) contained 850 drugs, and all were documented records on the medicinal uses of NPs [79]. The Greek physician Dioscorides (100 A.D.) recorded the collection, storage and uses of medicinal herbs. Monasteries in England, Ireland, France and Germany preserved the Western knowledge of natural medicines during the Dark and Middle Ages, whilst the Arabs preserved the Greco-Roman knowledge and expanded the uses of their own resources, together with Chinese and Indian herbs [79]. Avicenna, a Persian pharmacist, physician, philosopher and poet, contributed much to the science of pharmacy and medicine in the 8th century through his work titled “Canon Medicinae” [79].

The early years of the 20th century witnessed remarkable drug discoveries from natural sources. The world’s first broadly effective antibiotic substance, penicillin, was first isolated from the fungus *Penicillium notatum* by Sir Alexander Fleming, a Scottish physician and microbiologist, in 1928 [80]. This discovery led to the re-isolation and in vivo and clinical studies by Fleming and co-workers in the early 1940s and commercialization of synthetic penicillins, which ultimately revolutionized drug discovery and won them the 1945 Nobel Prize in Physiology and Medicine [81]. After the first clinical report on penicillin G, there was global interest in exploiting many natural resources for new bioactive natural products. The post-Fleming era witnessed the isolation of some sulphur-containing secondary metabolites (natural antibiotics) such as aztreonam, nocardicin G and imipenem from marine organisms [78]. Erythromycin was discovered from *Saccharopolyspora erythraea* as an antibacterial drug with a 14-membered macrocycle composed entirely of propionate units, broadly active against Gram-positive cocci and bacilli, and is used for mild to moderate upper and lower respiratory tract infections [82]. A typical natural product drug discovery process is represented as a flow chart in Figure 4.

Plant-derived NPs continue to contribute significantly to the discovery of newer drugs. For example, the bark of Pacific yew plant, *Taxus brevifolia*, gave taxol, a cytotoxic compound that was latter developed as Paclitaxel for the treatment of human lung cancer and other malignant tumours [83]. Some antiviral flavonoids such as 6-hydroxyluteolin 7-*O*-β-d-glucoside, nepitrin and homoplantaginin were isolated from the methanol extract of *Salvia plebeia*. These compounds were found to be active against influenza virus H1N1A/PR/9/34 neuraminidase [84]. Matteflavoside G from the rhizomes of *Onoclea struthiopteris* showed significant inhibitory activity against the H1N1 influenza virus neuraminidase with an EC_50_ value of 6.8 ± 1.1 μM and an SI value of 34.4 [85]. Extracts of *Humulus lupulus* and five South African medicinal plants, namely *Volkameria glabra*, *Cussonia spicata*, *Myrsine melanophloeos*, *Pittosporum viridiflorum* and *Tabernaemontana ventricose*, are known in traditional medicine to manage inflammatory and respiratory diseases such as the influenza virus, with quercetin and rutin identified among their putative active constituents [86] (Figure 5).

A review of high-throughput screening (HTS) by Novartis revealed that NPs were the most diverse compounds tested, with significantly higher hit rates compared to the compounds sourced from the synthetic and combinatorial libraries [87]. Some potential natural anti-TB agents include the antifungal phenazine and riminophenazine isolated from lichens. Clofazimine is a riminophenazine and TB drug originally discovered in 1954 through structural modifications of diploicin, extracted from *Buellia canescens*. It is currently used as a WHO group-five drug for multidrug resistant tuberculosis (MDR-TB) [88]. Natural products and NP-derived compounds, such as aztreonam, colistin, and tobramycin, have been developed for cystic fibrosis as inhalation drugs, while amikacin, arbekacin, and capreomycin are being developed for nontuberculous mycobacterial infection, bacterial pneumonia, and tuberculosis, respectively [89] (Figure 6).

For decades, many herbs and spices have been known to be used in folkloric medicines for the management of respiratory diseases. Some of these medicinal spices and herbs are now well established in modern medicine as dietary supplements, nutraceuticals, and whole drugs because of their identified and well-defined bioactive agents [90]. The peel methanol extract of *Opuntia ficus-indica,* known as the prickly pear cactus, has been reported to contain some in vitro anti-pneumonia compounds such as astragalin, quercetin 5,4′-dimethyl ether, isorhamnetin-3-O-glucoside and isorhamnetin [91]. Curcumins from the turmeric rhizomes (*Curcuma longa*) are known anti-inflammatory, antiviral, immune modulating, anti-lung cancer and anti-SARS-CoV-2 agents, as well as inhibitors of acute and chronic respiratory disorders [92,93]. Gingerols, 6-shogaol, zingerone, gingerenone-A, 6-dehydrogingerdione, β-bisabolene, α-curcumene and β-sequiphellandrene, all from the bulbs of ginger (*Zingiber officinale*), are known bioactive agents against asthma, inflammation, lung cancer, acute and chronic respiratory disorders, and respiratory viruses including coronaviruses [94]. 6-Gingerol was reported to decrease the gene expression and production of MUC5AC, through affecting the ERK- and p38 MAPK signalling pathways, thus inhibiting pro-inflammatory actions of many pulmonary diseases [95]. *Nigella sativa* L. (black cumin seeds) contain bioactive agents such as nigelline, thymol, thymoquinone, nigellidine, nigellicine, carvacrol, p-cymene, 4-terpineol, trans-anethol, α-pinene, α-hederin, and kaempferol-3-glucoside [96]. The constituents improved antioxidant enzymes (catalase, glutathione peroxidase and glutathione-*S*-transferase), and exhibit anti-inflammatory, immune modulatory and broncho-dilatory effects against obstructive RDs [96]. The inhibitory effect of nigellone on the release of histamine from mast cells has been implicated for the management of bronchitis and asthma [97].

Additionally, some NPs such as the anti-influenza ginkgetin, 4′-*O*-methylochnaflavone, hinokiflavone from *Ginkgo biloba*; six cinnamic amide alkaloids from *Tribulus terrestris* with considerable in silico SARS-CoV PLpro activity; procyanidin B1, procyanidin A2 and cinnamtannin B1 from the dried bark (cortex) of *Cinnamomum verum* with in vitro anti-SARS-CoV activity; and resveratrol and pterostilbene from grapes (*Vitis vinifera*) interfered with the SARS-CoV-2 infection cycle and significantly inhibited COVID-19 infection in primary human bronchial epithelial cells cultured under air–liquid interface conditions [9]. Likewise, some triterpenoids such as oleanolic acid, betulinic acid, ursolic acid and saikasaponins A, C, D, B1, B2, B3, and B4, which have been isolated from some medicinal plants, are known to exhibit significant antioxidant, anti-inflammatory, cytotoxic, antibacterial, antiviral and immune modulatory activities in lung diseases such as COPD, bronchitis, lung cancer, influenza and coronaviruses [9,95,98,99,100,101]. Some medicinal plants and their active ingredients are presented in Table 2, while the structures of some of these bioactive compounds are shown in Figure 7, Figure 8, Figure 9, Figure 10 and Figure 11.

**Table 2 molecules-27-03054-t002:** Some plant-derived natural products and their biological potentials against common respiratory diseases.

Natural Source (Family)	Medicinal Use	Biological Property	Active Part/Ingredient	Reference
*Aerva lanata* (L.) A. L. Juss. ex Schultes. (Amaranthaceae)	Asthma, cough	Anti-asthmatic, antimicrobial, immunomodulatory, diuretic, anti-inflammatory	Aerial part ethanol extract	[102]
*Ageratum conyzoides* L. (Asteraceae)	Asthma	Antihistaminic, anticataleptic	Leaf hydroalcoholic extract	[103]
*Allium sativum* L. (Amaryllidaceae)	Asthma, pneumonia, influenza, COVID-19	Antibacterial, antifungal, anti-inflammatory, immune modulating, antiviral (SARS-CoV-2), anticancer	Organosulfur compounds such as diallyl thiosulfinate (allicin) and diallyl polysulfane from the bulb	[104,105,106]
*Amburana cearensis* A.C. Smith (Fabaceae)	Asthma and other respiratory diseases	Anti-asthmatic	flavonoid-5,7,4′-trihydroxy-3-methoxyflavone (isokaempferide) from the trunk bark extract	[107]
*Angelica keiskei* (Miq.) Koidz. (Apiaceae)	COVID-19	Antiviral, anti-SARS-CoV-2	xanthoangelol E from the leaf extract	[108]
*Argemone mexicana* L. (Papaveraceae)	Asthma	Antiallergic, antistress	Stem aqueous extract	[109]
*Asystasia gangetica* (L.) T. Anderson (Acanthaceae)	Asthma	Anti-asthmatic, anti-inflammatory	Leaf fractions (hexane, ethyl acetate, methanol)	[110]
*Cinnamon zeylanicum* Blume (Lauraceae)	COPD, lung cancer, flu-related diseases such as influenza and coronaviruses	Antioxidant, anti-inflammatory, antimicrobial, fungitoxicant against respiratory tract mycoses (*Candida* sp.), antiviral, anticancer, immune modulatory	Cinnamaldehyde and trans-cinnamaldehyde, procyanidins, catechins, volatile oils from the bark	[111,112,113,114]
*Cassia sophera* (L.) Roxb. (Fabaceae)	Asthma, bronchitis (India)	Anti-asthmatic	Leaf fractions (Chloroform, ethyl acetate, methanol)	[115]
*Chamaecyparis obtusa var. formosana* Hayata (Siebold & Zucc.) Siebold & Zucc. ex Endl. (Cupressaceae)	COVID-19, SARS-CoV	Anti-SARS-CoV-2, cytotoxic	Ferruginol, betulonic acid, betulinic acid, savinin, from the heartwood extract	[108]
*Citrus limon* (L.) Burm. and *Citrus* peel and fruit (Rutaceae)	Bronchitis, Flu-related illnesses, COPD, lung cancer	Antioxidant, anti-inflammatory, immune modulatory, antibacterial, antiviral (rhinovirus, influenza virus, coronaviruses), anticancer	Flavonoids such as eriocitrin, hesperidin or diosmin, apigenin, naringin, naringenin, narirutin, quercetin, luteolin, hesperetin, nobiletin	[116,117,118,119,120,121]
*Crinum glaucum* A Chev. (Amaryllidaceae)	Asthma, cough, convulsion, oral thrush, COPD, lung	Antiallergic, antifungal (candidacidal),	Bulb aqueous extract	[122,123,124]
*Cryptomeria japonica* (Thunb. ex L.f.) D.Do. (Cupressaceae)	COVID-19	Antiviral, anti-SARS-CoV-2	7β-hydroxydeoxy- cryptojaponol from the heartwood extract	[108]
*Curcuma longa* L. (Zingiberaceae)	Common cold, COVID-19, pneumonia, influenza, bronchial asthma, COPD, lung cancer	Antiviral, anti-SARS-CoV infections, cytotoxic, anti-influenza virus, immune modulating, anti-inflammatory	Curcumins and turmerones from the rhizomes (roots)	[92,93,125,126,127,128,129,130]
*Euphorbia hirta* Linn. (Euphorbiaceae)	Asthma, bronchitis, hay fever, oral thrush	Antihistaminic, antiallergic, anti-anaphylactic, antibacterial, antifungal, anti-inflammatory	Quercitrin, rutin, borneol, quercitol, euphorbin, gallic acid from the aerial part ethanolic extract	[131,132]
*Ficus deltoidea* Jack (Moraceae)	Flu including COVID-19	Antiviral (coronaviruses)	Rhoifolin from the leaf extract	[108]
*Hedera helix* L. (Araliaceae)	COPD, COVID-19, bronchial asthma, bronchitis	Anti-inflammatory, expectorate, antiviral	Hederasaponin-C, hederagenin and α-hederin from the leaf extract	[133]
*Lycoris radiata* (L′Héritier) Herbert (Amaryllidaceae)	Flu such as SARS-CoV infections, COVID-19	Antiviral, anti-SARS-CoV, anti-SARS-CoV-2	Lycorine from the stem cortex extract	[134]
*Mentha spicata* L. (Lamiaceae)	Asthma (Japan)	Antihistaminic	Sideritiflavone from the leaf methanol extract	[135]
*Momordica dioica* Roxb. ex. Willd (Cucurbitaceae)	Asthma, bronchitis (India)	Antihistaminic	Pulp methanol and aqueous extracts	[136]
*Myrica esculenta* Buch. (Myricaceae)	Asthma, bronchitis (India)	Antiallergic, anti-inflammatory, bronchodilator, anti-anaphylactic	Aerial part and stem bark ethanol extracts	[137]
*Nigella sativa* L. (Ranunculaceae)	Bronchitis, COPD, pneumonia, flu, lung cancer	Antioxidant, immune modulatory, anti-inflammatory, preventive effect in respiratory disorders, broncho-dilatory, cytotoxic	Thymoquinone, nigellone, thymol, carvacrol, p-cymene, 4-terpineol, trans-anethole, α-pinene, α-hederin, kaempferol glucoside	[96,138]
*Panax ginseng* C. A. Meyer (Araliaceae)	Oral thrush, acute respiratory illness (pharyngitis, bronchitis, COPD, respiratory tract infections	Antifungal (candidacidal), anti-inflammatory, antibacterial, antiviral (rhinovirus, respiratory syncytial virus, coronaviruses),	Ginsenosides such as 20(S)-protopanaxatriol and 20(S)-protopanaxadiol from the root extract	[139,140,141]
*Piper betel* Linn.	Asthma, cold, cough (Asia and Africa)	Anti-asthmatic	Leaf ethanol and aqueous extracts	[142]
Polyherbal formulations containing some medicinal herbs and spices	Different respiratory diseases including asthma	Anti-asthmatic, mast cell stabilization, anti-inflammatory, anti-spasmodic, antiallergic, anti-anaphylactic, immunomodulatory and inhibition of mediators such as leukotrienes, histamine, cytokines	Polyherbal mixture	[143]
*Terminalia chebula* Retz.	Respiratory syncytial virus	Broad spectrum Antiviral	Chebulagic acid, punicalagin	[144]
*Rheum palmatum* L. (Polygonaceae)	COVID-19 disease	SARS-CoV-2 inhibition	Chloroform fraction from ethyl acetate and 75% ethanolic extract	[145]
*Salvia miltiorrhiza* Bge (Lamiaceae)	COPD	Antioxidant, anti-inflammatory	Tanshinone IIA from the root extract	[146]
*Thymus vulgaris* L. (Lamiaceae)	Pertussis, bronchitis, asthma, acute lung injury, influenza, COVID-19	Antioxidant, antimicrobial, anti-inflammatory, antiviral (influenza, coronaviruses)	Thymol, p-cymene, linalool, carvacrol from the leaf infusion	[147,148,149,150]
*Tylophora indica* (Burm. f.) Merr. (Apocynaceae)	Coronavirus infections	Anti-SARS-CoV	Tylophorinine	[151]
*Zingiber officinale* Roscoe (Zingiberaceae)	Asthma, COPD, common cold, bronchitis, influenza, coronaviruses, lung cancer	Antioxidant, anti-inflammatory, antiviral (SARS-CoV), immune modulatory, cytotoxic	6-Gingerol, 8-gingerol, 10-gingerol and 6-shogaol from the bulbs	[94,152]

## 5. Structure–Activity Relationships of Some Promising Natural Products against Common Respiratory Diseases

The study of the structure–activity relationships (SARs) is an approach designed to find the relationships between chemical structures of ligands and biological targets of studied compounds [153]. It has become increasingly essential as a tool for organizing, mining, and interpreting data, to guide further investigation for drug discovery [154]. It is also a strategy to increase the value of the activity initially detected [155]. Natural products contain steric and electronic features in their bioactive sites (pharmacophores), which are responsible for the optimal supramolecular interactions with specific biologic targets and to trigger (or block) their biologic responses. The most used features for describing pharmacophore sites are hydrogen bond acceptors and donors, acidic and basic functional groups, aliphatic and lipophilic moieties, aromatic- and hydroxyl-hydrophilic moieties amongst others [156].

Reports have shown the SARs of some natural products implicated against some common respiratory diseases (Table 3). Pires et al. [157] reported the influence of methyl-, hydroxyl-, and carbonyl functional groups on the in vitro anti-TB activities of eight coumarin derivatives from *Calophyllum brasiliense*, with MIC ranging from 15.6–62.5 μg/mL and a cytotoxicity range of 4.5–82.0 μg/mL against *Mycobacterium tuberculosis* H37Rv and its multidrug-resistant clinical isolates. Here, the carbonyl and hydroxyl groups enhance anti-*M. tuberculosis* activity by the inhibition of acid-fastness formation in the mycobacterial cell wall, while the presence of lipophilic side chains such as the alkyl substituent at the C-3 position and the presence of double bonds increase the lipophilicity of the compound, thus, helping it to penetrate the lipid-enriched mycobacterial cell wall [157] (Figure 12).

Quercetin 5,4′-dimethyl ether isolated from the fruit peel methanol extract of *Opuntia ficus-indica* has been reported to demonstrate double-fold in vitro anti-pneumonia activities, with MIC values of 0.49 and 0.98 µM against *Klebsiella pneumonia* and *Moraxella catarrhalis*, respectively, when compared to Imipenem, a standard anti-pneumonia drug [91]. Furthermore, an in silico molecular docking study of the compound revealed high H-bonding affinity with key amino acids such as threonine, asparagine, and tyrosine, thus suggesting it to be a natural quorum-sensing inhibitor, which is a key anti-pneumonia property [94]. Wollamide B isolated from *Streptomyces nov. sp.* (MST-115088) has been reported to show considerable in vitro anti-TB activity with an IC_50_ value of 3.1 μM against *Mycobacterium bovis*. The presence of the basic amino acid ornithine and clusters of lipophilic amino acids was shown to significantly contribute the typical cationicity and amphiphilicity to the molecule [158].

The presence of *C*- and *N*-glycosylation has also been reported to enhance in vitro anti-TB activity. For instance, the *C*-glycosylated benz[α]anthraquinone derivatives and an *N*-glycosylated arenimycin isolated from *Streptomyces* species and *Salinispora arenicola*, respectively, showed strong activity against *Mycobacterium tuberculosis* within an MIC range of 5.88–24.32 µM, while the latter molecule exhibited an MIC value of 1.5 µM [159] (Figure 12). It is also noteworthy that unsaturation in the C ring (∆^2^), the number and position of hydroxyl groups at the A and B rings, and the carbonyl group at C-4 of ring C for natural flavonoids are reported to contribute considerably to their anti-inflammatory properties in some common lung diseases [160].

**Table 3 molecules-27-03054-t003:** Structure–activity relationships of some natural products against common respiratory diseases.

Compound	Source	Bioactivity	SAR	Reference
4-Deoxybostrycin	*Alternaria eichhorniae* 5 (isolate Ae5)	In vitro anti-mycobacterial (TB) activity (IC_50_ of 12.5 µM), better inhibitory effect on clinical multidrug-resistant *M. tuberculosis* (K2903531 and 0907961) than (Nigrosporin, the standard antitubercular drug	Hydroxyl group at C-5 enhances binding effect between the bacteria active site and the molecule methyl at C-7 increases lipophilicity and transport across bacterial cell membrane	[159]
Cleistrioside-2	*Cleistopholis patens* (Benth.) Engl. & Diels C. glauca Pierre ex Engl. & Diels	Cytotoxicty against the human lung cancer cell lines (NCI-H460) at CC_50_ = 9.1 µM. Antibacterial activity against *Streptococcus pneumonia* at MIC = 4 µM	C-3 glycosylation and C-4 acetate group on the terminal sugar	[161]
Ginsenosides	*Panax ginseng* C. A. Meyer	Antioxidant, immune-modulatory, and antiviral properties	Presence of an aglycon, protopanaxadiol, and a part of the sugars may contribute to the immune-modulatory properties of the herbs	[162]
Hydroquinone	*Albizia coriaria* Welw ex. Oliver	In silico anti-TB activity. Better binding affinities (−7.8 kcal/mol) for the mycobacterial ATPase and polyketide synthase-13 than isoniazid and rifampicin	Interactions of the co-crystalized ligand with amino acid residues in the binding site of ATP synthase	[163]
Liquiritin apioside	*Glycyrrhiza glabra* L. *Paeonia lactiflora* Pall.	Chronic obstructive pulmonary disorder (COPD)	Presence of hydroxyl group at C-5 and C-7 of ring A promotes enzymatic oxidation and consequently bonding of flavonoids with biomacromolecules	[164]
Jusan coumarin (Dicoumarin)	*Artemisia glauca* Pall. ex Willd	In silico anti-SARS-CoV-2 activity	Presence of pharmacophoric features such as two H-bond donors, one H-bond acceptor, an aromatic ring and two hydrophobic centres	[165]
Ophiobolin K	*Emericella variecolor* IFM42010	Anti-tubercular activity.	Configuration of C-6 is key for optimal activity	[159]
Theopederin	*Theonella* species such as *Theonella swinhoei* Gray	In vitro antiviral activity against SARS-CoV-2)	Inhibition of SARS-CoV-2 main protease aided by terminal guanidine, cyclic hemiacetal linkage, and the length of the side chain	[166]
Quercetin	Many higher plants including *Polygoni avicularis* Herba	Anti-inflammatory effects against lung diseases such as asthma, allergy, and acute respiratory diseases, and chronic respiratory disorder (COPD)	The presence of ketonic carbonyl and double bond at C-2/C-3 of ring C induces coplanarity between rings A and C, favouring the interaction of the flavonoid with the enzymatic site receptor. Hydroxyl group at C-5/C-7 of ring A as well as at the C-3′ and C-4′ of ring B favours enzymatic and consequently bonding of flavonoids with biomacromolecules	[160,167]
Quercetin 5,4′-dimethyl ether	*Opuntia ficus-indica* (L.) Miller	Higher in vitro anti-pneumonia than Imipenem. In silico quorum sensing efficacy	π–π interaction involving flavone A- and C-rings, π–alkyl interactions involving A-, B- and C-rings	[91]
Vernogratioside A & B	*Vernonia gratiosa* Hance	In silico anti-SARS-CoV-2 main protease with comparable −7.2 and −7.6 kcal/mol binding affinity to N3 inhibitor (−7.5kcal/mol)	C-3 glycosylation and presence of alkyl substituent	[168]
Wollamide B	*Streptomyces nov.* sp. (MST-115088)	In vitro anti-tubercular activity	The presence of the basic amino acid ornithine and clusters of lipophilic amino acids impart the typical cationicity and amphiphilicity to the molecule	[158]

**Figure 12 molecules-27-03054-f012:**
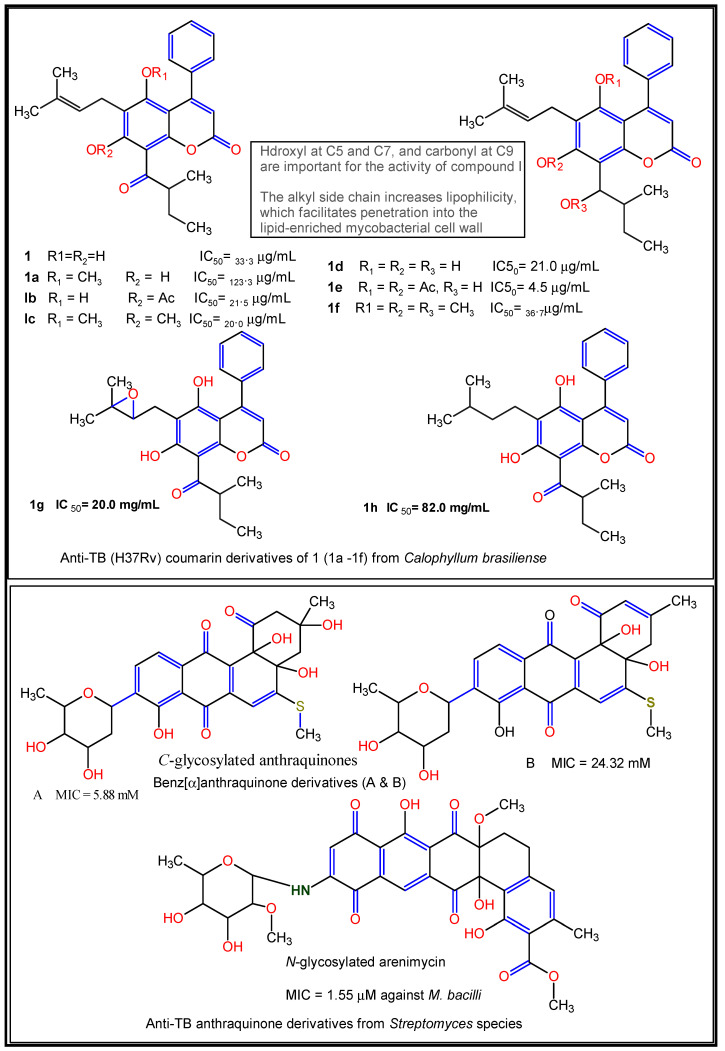
Structure–activity relationships of some natural product-derived anti-TB compounds. Active moieties indicated in coloured forms [157,159].

## 6. Mechanisms of Action of Some Plant-Derived Lead Compounds against Common Respiratory Diseases

Often, the human respiratory tracts become liable to inflammation upon microbial (especially bacterial and viral) infection and physical injury [169]. Respiratory inflammation is a hallmark of many respiratory diseases, which include asthma, COPD, and acute respiratory disorders (ARDs) [170]. During the inflammation process, inflammatory cells, which include eosinophils, lymphocytes, and macrophages, are activated to serve as the sources of different inflammatory mediators such as histamine, interleukins (IL-4, IL-1β, IL-6, and IL-5), leukotriene, prostaglandins, nitric oxide, and tumour necrosis factor (TNF-α) [171]. The release of these inflammatory mediators causes several abnormalities in the lungs and their function [170,171]. Therefore, natural products (NPs) that can target the epithelial–mesenchymal transition (EMT), oxidative stress, fibroblast activation, inflammatory injury, metabolic regulation, and extracellular matrix accumulation within the respiratory tracts are regarded as candidate anti-inflammatory agents that can be optimized as leads for new respiratory drugs [172]. The basic mechanism of action of the chemical agents involves regulating redox status, inhibiting the activities of bacteria and viruses, regulating the protease/anti-protease balance, blocking the NF-κB and MAPK signalling pathways, inhibiting the production of cytokines, suppressing the activation and migration of inflammatory cells, inhibiting the synthesis and activation of adhesion factors and growth factors, controlling the cAMP-PKA and PI3K/Akt signalling pathways, and increasing TIMP-1 expression to serve as anti-inflammatory agents in the lungs [171,173] (Figure 13).

The anti-inflammatory properties of curcumins against lung inflammation induced *Klebsiella pneumonia* has been demonstrated in a mouse model experiment [174]. Here, it was shown that curcumin ameliorates lung inflammation considerably by decreasing the bacterial load in the lung tissue and inducing a significant decrease in neutrophil influx into the lungs as well as in the production of MDA, NO, and MPO activity and TNF-alpha levels, whereas Augmentin, the standard antibiotic, takes care of bacterial proliferation. The study, however, highlighted the potential usefulness of curcumin as an adjunct therapy along with some antibiotics as an anti-inflammatory or an immunomodulatory agent in the case of acute lung infection [174]. Studies have also shown the significant anti-inflammatory actions of quercetin, which are believed to be mediated through the inhibition of phospholipase A2 (via arachidonic acid), lipoxygenase, cyclooxygenase, and thromboxane enzymes and through the modulation of iNOS, thereby inhibiting NO production [175,176].

Some alkylated chalcones such as xanthoangelol A-G, isolated from *Angelica keiskei* leaves, have been reported to show considerable in vitro and in silico anti-SARS-CoV activity. Xanthoangelol D particularly inhibited the SARS-CoV cysteine proteases with an IC_50_ of 1.2 µM, by both competitive and non-competitive modes, thus suggesting the molecule to be a candidate protease inhibitor in SARS-CoV-related infections [177]. Similarly, some natural phenolic compounds such as brazilin, theaflavin-3,3′-digallate and curcumin have been reported to have remarkable in vitro anti-SARS-CoV-2 activity with an IC_50_ ≥ 10 µM, among 56 polyphenolic compounds and plant extracts that were tested. The compounds were said to bind with the receptor-binding domain of SARS-CoV-2 spike protein, thus significantly inhibiting viral attachment to the human angiotensin-converting enzyme 2 receptor and cellular entry of pseudo-typed SARS-CoV-2 virions [178]. 

Tetragalloyl quinic acid isolated from *Galphimia glauca* has been reported as an in vivo anti-asthmatic agent at 5 mg/kg orally, by suppressing allergen- and platelet-activating factor, PAF-induced bronchial obstruction, PAF-induced bronchial hyperreactivity, and thromboxane biosynthesis in vitro. Androsin from *Picrorhiza kurroa* also demonstrated similar in vivo activity at 10 mg/kg orally (0.5 mg inhalative) by preventing allergen- and PAF-induced bronchial obstruction [179].

Additionally, ganoderic acid C1 in the ASHMI^TM^ herbal formula comprising *Ganoderma lucidum Sophora flavescens* and *Glycyrrhiza uralensis* has been reported to have potential for treating TNF-α mediated inflammation in asthma and other inflammatory diseases [180]. Based on clinical studies, the herbal formula significantly reduced TNF-α production by murine macrophages (RAW 264.7 cells) and peripheral blood mononuclear cells (PBMCs) from asthma patients [180,181]. The inhibition was associated with down-regulation of NF-κB expression and partial suppression of MAPK and AP-1 signalling pathways [181].

The combination of synephrine and stachydrine, both alkaloids from the dried rind of ripe *Citrus reticulata* Blancon fruits, has been reported to show significant spasmolytic effects on acetylcholine chloride (ACh)-induced contractions in isolated guinea pig trachea by activating β-2 adrenergic receptor signalling [182]. They also showed synergistic protection against histamine-induced experimental asthma by prolonging the latent period. Stachydrine acts as the antitussive component and is capable of significantly reducing citric acid-induced coughing; thus, the broncho-dilatory and antitussive effects of the combined alkaloids might explain their use in traditional Chinese medicine as an anti-asthmatic remedy [182]. The mechanisms of action of some plant-derived compounds implicated against some common respiratory diseases are shown in Table 4.

**Table 4 molecules-27-03054-t004:** Mechanisms of action of some plant-derived lead compounds against some common respiratory diseases.

Natural Products	Source	Mode of Action/Biological Effect	References
1,8-Cineol	*Eucalyptus globulus* ssp. *globulus* Labill.	Inhibits nuclear translocation of NF-κB p65 and NF-κB-dependent transcriptional activity. Anti-asthmatic properties	[183,184]
3-Methoxycatalposide	*Platylobium rotundum* I.Thomps.	Inhibits the expression of cyclooxygenase (COX)-2, nitric oxide synthase (iNOS), and proinflammatory genes (IL-6, IL-1β, and TNF-α). Anti-asthmatic properties	[184]
3-O-α-L-rhamnopyranosyl-(1→2)-β-D-xylopyranosyl-(1→2)-β-D-xylopyranosyl-21-cinnamoyloxyoleanolic acid	*Burkea africana* Hook	Inhibition of the viral surface protein neuraminidase. In vitro anti-influenza virus activity, IC_50_ = 0.05 µM	[185]
3-O-α-L-rhamnopyranosyl-(1→2)-β-D-xylopyranosyl-(1→2)-[α-l-rhamnopyranosyl-(1→4)]-β-D-xylopyranosyl-21-cinnamoyloxyoleanolic acid	*Burkea africana* Hook	inhibition of the viral surface protein neuraminidase. In vitro anti-influenza virus activity, IC_50_ = 0.17 µM	[185]
4-(α-L-rhamnopyranosyloxy) benzyl isothiocyanate	*Moringa oleifera* Lam.	Inhibits inflammatory responses such as eosinophils, macrophages, dendritic cells, T-helper type 2 (Th2) cells, IgE-secreting B cells and mast cells accumulation. Anti-asthmatic activity. EC_50_ ≤ 50 mM in histamine and acetylcholine-exposed guinea pig ileum	[186]
4-(β-D-glucopyranosyl-1→4-α-L-rhamnopyranosyloxy)-benzyl thiocarboxamide	*Moringa oleifera* Lam.	Inhibits inflammatory responses such as eosinophils, macrophages, dendritic cells, T-helper type 2 (Th2) cells, IgE-secreting B cells and mast cells accumulation. Anti-asthmatic activity. EC_50_ ≤100 mM in histamine and acetylcholine-exposed guinea pig ileum	[186]
Caffeic acid	*Echinacea purpurea* (L.) Moench	Inhibits in vitro SARS-CoV helicase activity, IC_50_ = 0.1 μM	[187]
Chlorogenic acid	*Echinacea purpurea* (L.) Moench	Inhibits angiotensin converting enzyme (ACE), IC_50_ = 0.1 μM. In vitro anti-SARS-CoV	[187]
Cleistanthin A	*Cleistanthus**collinus* (Roxb.) Benth. ex Hook. f.	Inhibits the endocytic machinery, that is, by inhibiting V-type ATPase and elevating endolysosomal pH (EC of 0.1 μM). Anti-SARS-CoV	[188,189]
Cleistanthoside A tetraacetate	*Phyllanthus taxodiifolius* Beille	Neutralizes endolysosomal acidity and decreases the activity of V-type ATPase with an EC_50_ of 50 nM. Anti-SARS-CoV activity	[188]
Cryptotanshinone	*Salvia miltiorrhiza* Bunge	Inhibition of the in vitro SARS-CoV PLpro, IC_50_ = 0.8 μM	[190]
Curcumin	*Curcuma longa* L.	Inhibits SARS-CoV PLpro, IC_50_ = 5.7 μM, in vitro Ameliorates pneumonia-induced lung injury through a reduction of the activity and infiltration of neutrophils and the inhibition of inflammatory response in pre-clinical pneumonia models. Inhibits the production of MDA, NO, MPO activity and TNF-alpha levels. Prevents lung infections	[174,191,192]
(+)-Hopeaphenol	*Ampelopsis brevipedunculata var. hancei* (Planch.) Rehder & Li.	Inhibition of the SARS-CoV helicase activity, IC_50_ = 1.6 μM	[193]
Matteflavoside G	*Matteuccia struthiopteris* (L.) Tod	Inhibits H1N1 influenza virus neuraminidase with an EC50 of 6.8 μM and an SI value of 34.4	[9]
Methyl galbanate	*Ferula assa-foetida* L.	Inhibits H1N1 influenza virus with an in vitro IC_50_ of 0.26 μM and significant in vitro cytotoxicity against human liver and lungs cancer cells	[194]
Scutellarein	*Scutellaria baicalensis* Georgi	Inhibition of the SARS-CoV helicase activity, IC_50_ of 0.86 μM	[195]
Silvestrol	*Aglaia odorata* Lour.	Inhibits the replication of MERS-CoV with an EC50 of 1.3 nM. acting as an inhibitor of RNA helicase eIF4A and protein expression via blocking replication/transcription complex formation	[196]
Tetra-O-galloyl-D-glucose	*Rhus chinensis* Mill.	Inhibition of the SARS-CoV S protein-ACE2 interaction, IC_50_ = 10.6 μM	[197]
(+)-Vitisin A	*Ampelopsis brevipedunculata var. hancei* (Planch.) Rehder & Li.	Inhibitory action against angiotensin converting enzyme (ACE), IC_50_ = 1.5 μM	[193]
Xanthoangelol E	*Angelica keiskei* (Miq.) Koidz	Inhibition of the SARS-CoV PLpro activity, IC_50_ = 1.2 μM	[177]

## 7. Conclusions and Future Prospects

Medicinal plants and other natural sources continue to provide man with scaffolds of chemically unique and biologically active agents as drug candidates against common respiratory diseases (RDs) [193]. These lead agents include colchicine, curcumin, turmerones, gingerols, forsythiaside A, glycyrrhizin, mangiferin, zingerone and many other plant-derived NPs that have been mentioned in this review [193]. However, despite their promising biological activities, extensive clinical and toxicological reports that validate their clinical efficacy and safety are still lacking.

Therefore, future study should focus on the isolation and identification of non-/less toxic and more effective natural compounds through bioassay-guided isolation, high-throughput-screenings, metabolomics, molecular modelling, virtual screening, natural product libraries, and database mining [198], while elucidation of the mechanisms of action of bioactive compounds, lead optimization, toxicological considerations, product formulation, evaluation of pharmacokinetic parameters, dosage regimens, and targeted drug delivery will remain crucial in the discovery and development of respiratory drugs [77,199]. The development of new-age technologies such as the application of biogenetic metal-based nanoparticles in medicine (nanomedicine) is another giant stride, which so far has ensured successful drug delivery with the use of nebulizers that create a particle size capable of reaching the alveoli, for faster and more effective treatment at low therapeutic doses [200,201].

Perhaps the next generation of lead compounds that would effectively manage many of the respiratory problems of mankind lies under our very nose, demanding more intensive scientific investigations. “Respiratory disease on the rise, natural products to the rescue”.

## Figures and Tables

**Figure 3 molecules-27-03054-f003:**
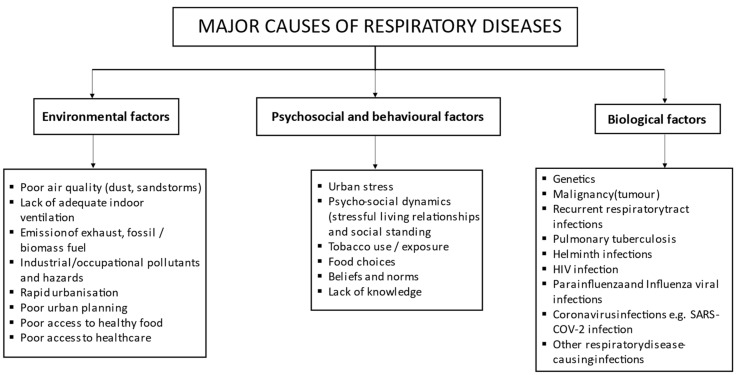
Major causes of respiratory diseases, adapted with permission from Ku et al. [31].

**Figure 4 molecules-27-03054-f004:**
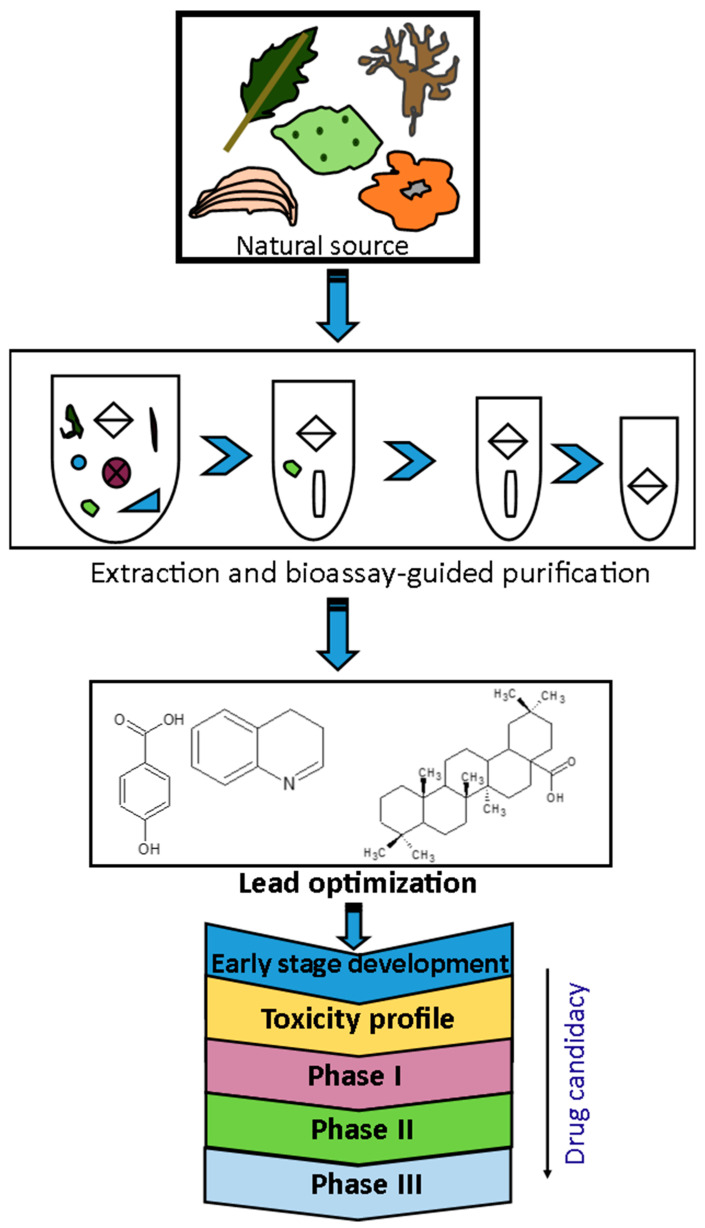
Stages involved in the identification of new drugs from natural sources. Adapted with permission from Quan et al. [55] (Copyright 2016, Elsevier Publisher).

**Figure 5 molecules-27-03054-f005:**
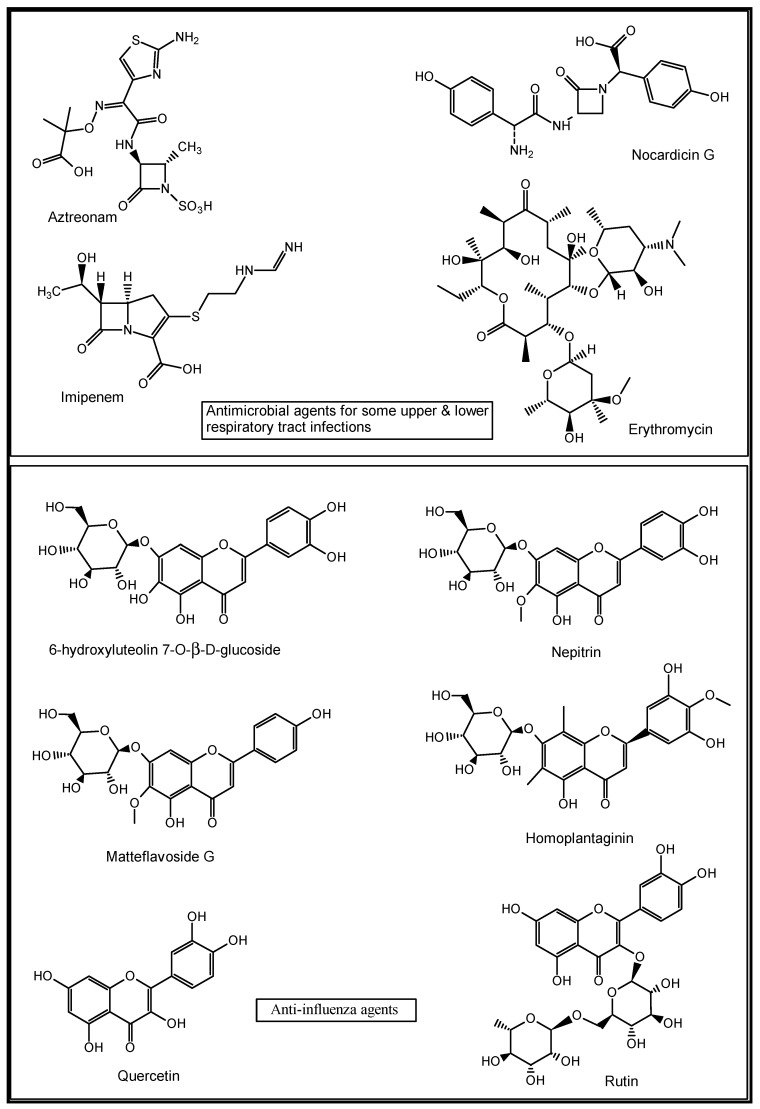
Structures of some natural anti-infective and anti-influenza agents [78,82,84,85,86].

**Figure 6 molecules-27-03054-f006:**
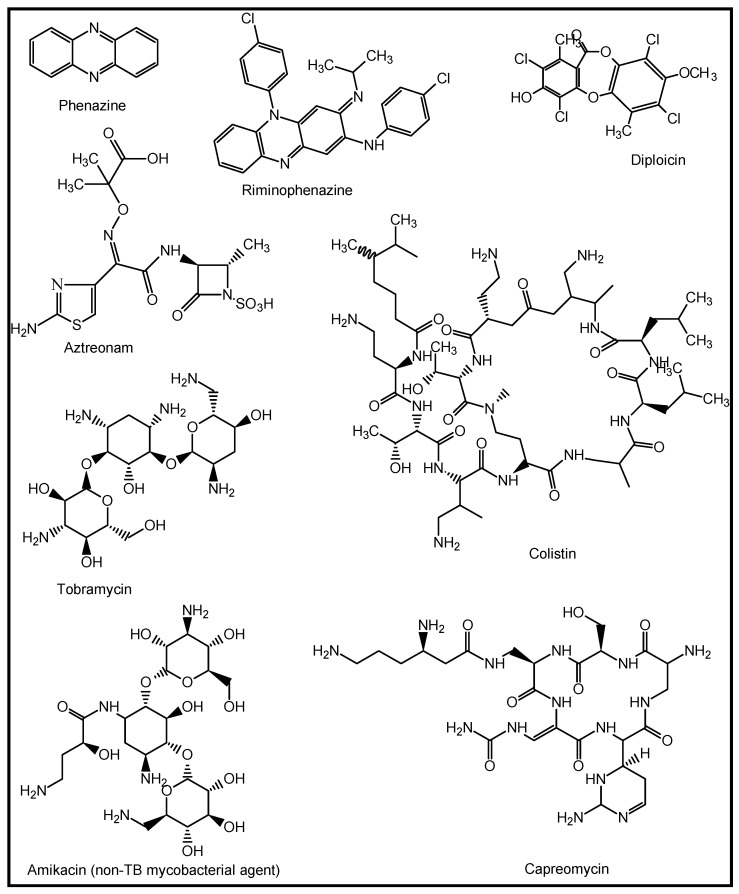
Structures of some nature-inspired agents against tubercular (TB) and non-TB mycobacterial infections [88,89].

**Figure 7 molecules-27-03054-f007:**
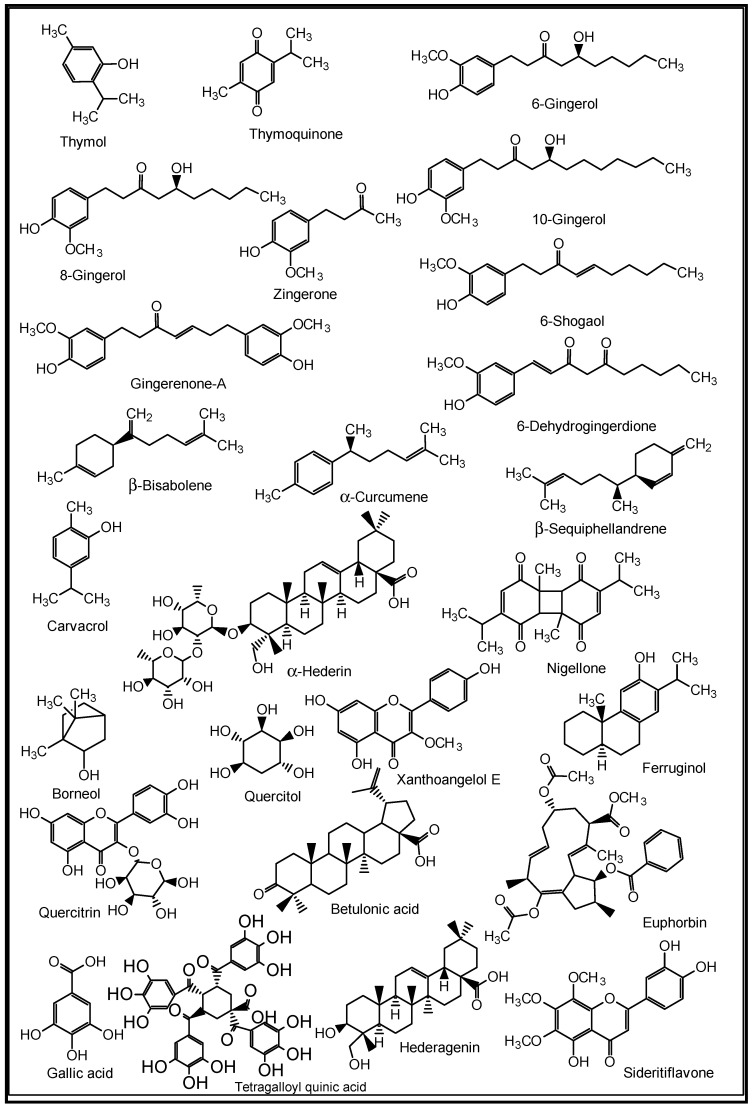
Some plant-derived anti-asthmatic compounds.

**Figure 8 molecules-27-03054-f008:**
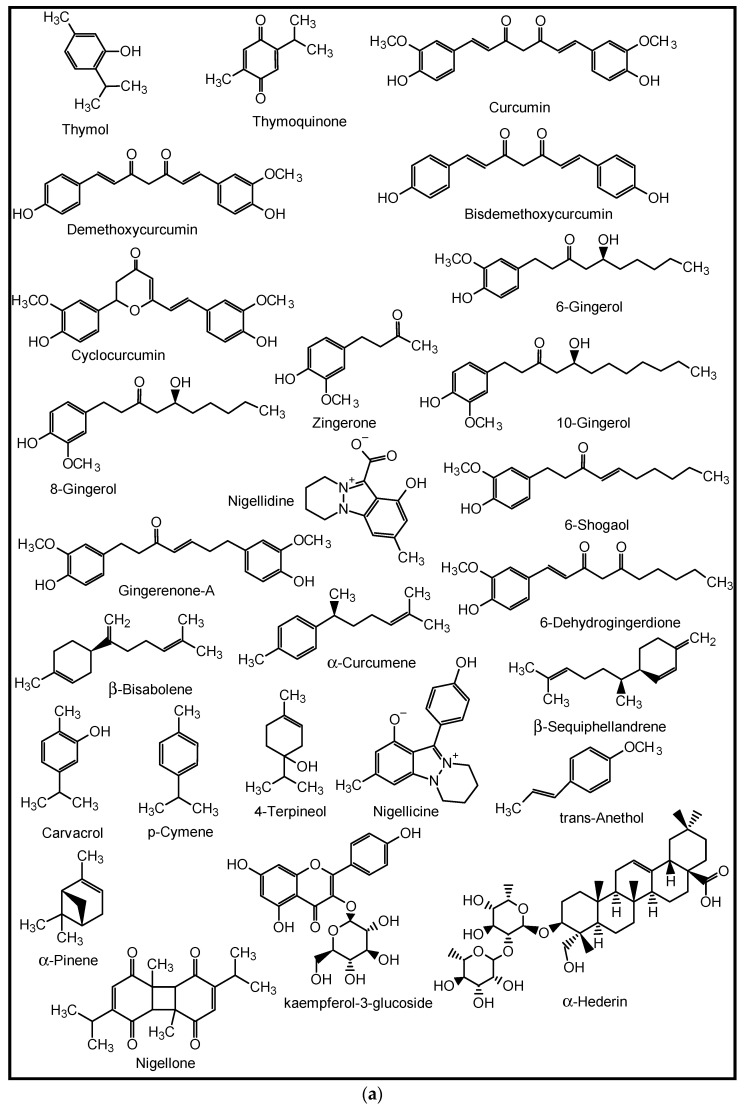
(**a**,**b**) Some plant-derived anti-infective (COPD) compounds.

**Figure 9 molecules-27-03054-f009:**
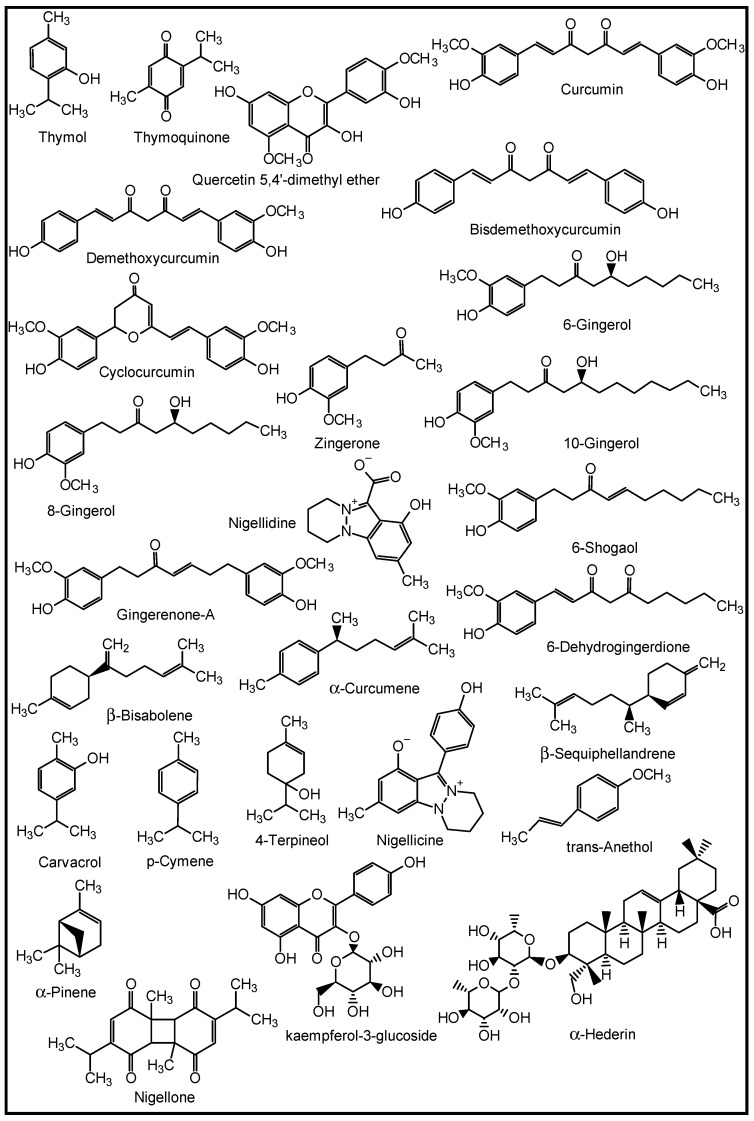
Some plant-derived anti-pneumonia compounds.

**Figure 10 molecules-27-03054-f010:**
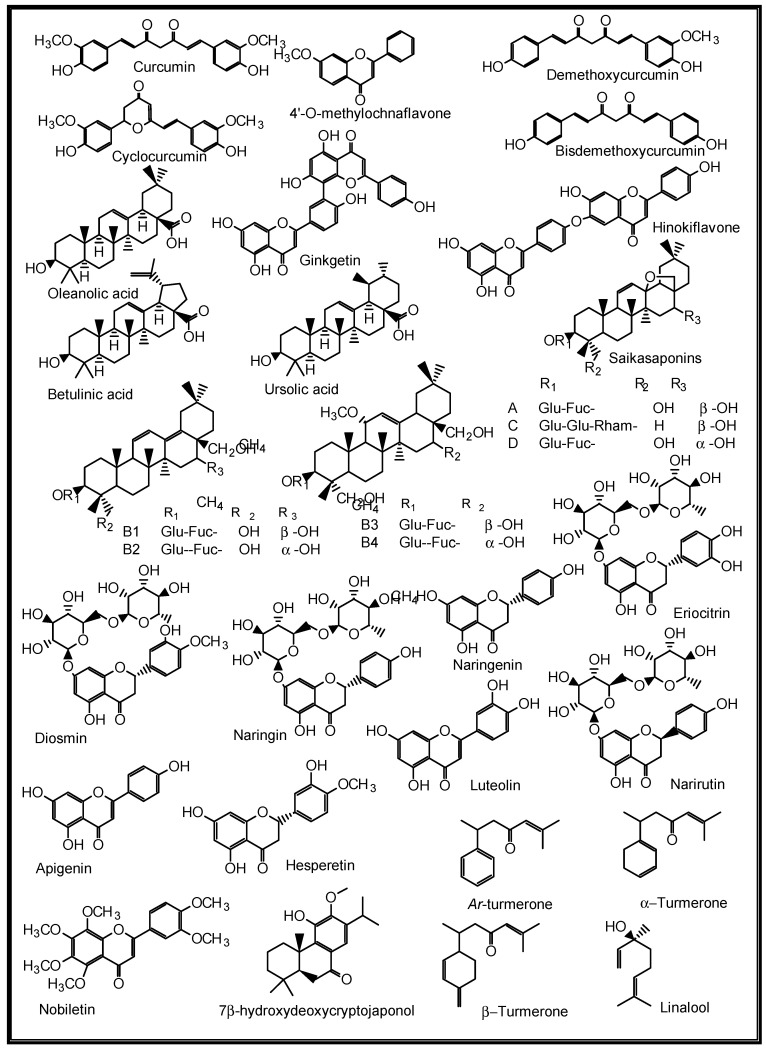
Some plant-derived anti-influenza compounds.

**Figure 11 molecules-27-03054-f011:**
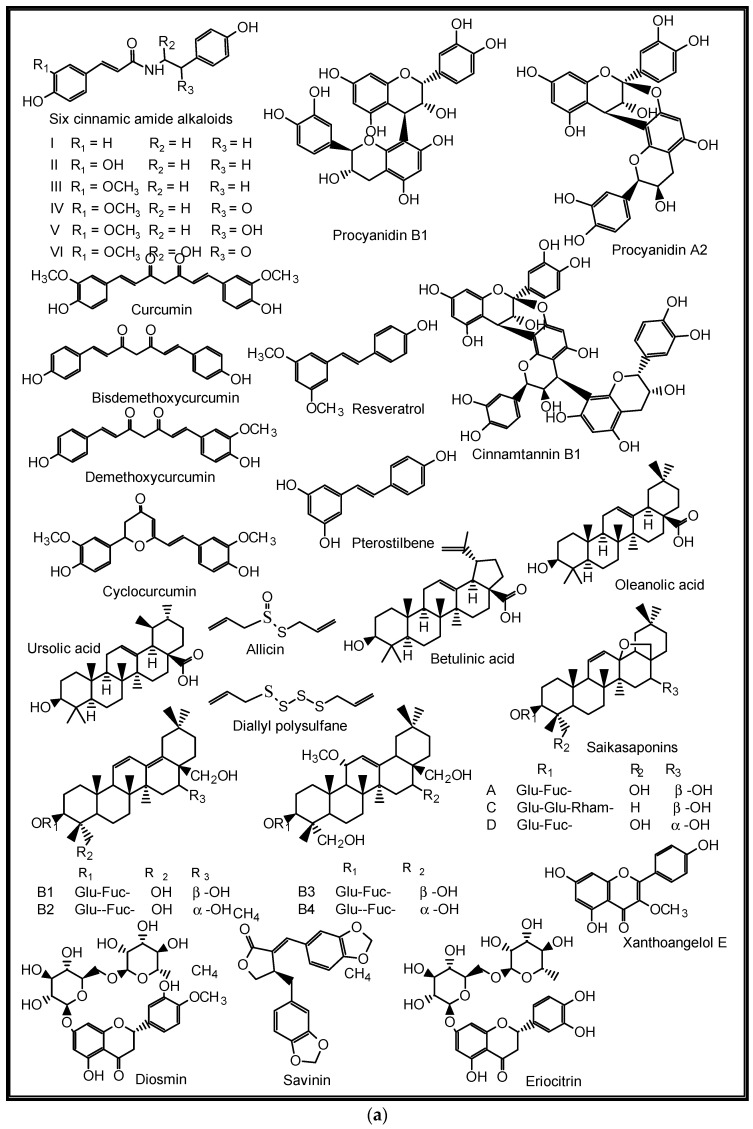
(**a**,**b**) Some plant-derived anti-SARS-CoV-2 compounds.

**Figure 13 molecules-27-03054-f013:**
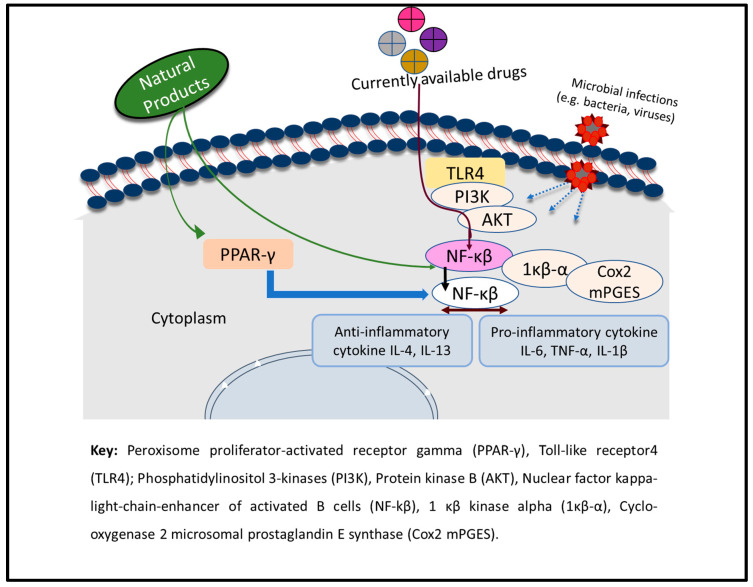
Mechanism of action of natural products during inflammation-induced respiratory illness. Adapted with permission from Timalsina et al. [171] (Copyright 2021, Hindawi).

**Table 1 molecules-27-03054-t001:** Common respiratory diseases, their levels of occurrence, and causes.

Respiratory Disease	Global Morbidity Ranking [33,34,35,36]	Cause	Class	Reference
COPD	1st	*Haemophilus influenza* and *Streptococcus pneumoniae*	Bacteria	[37]
		Influenza viruses, Rhinoviruses	Virus	
Asthma	2nd	Allergens and irritants (pollen, mould, dust, feathers, animal fur, smoke, fumes, perfume) Psychosocial (mental stress, laughter) Medicines (ibuprofen, aspirin)	-	[38]
		*Haemophilus influenza* and *Streptococcus pneumoniae*	Bacteria	
Pulmonary hypertension	3rd	High blood pressure, cirrhosis, congenital and coronary heart diseases, emphysema, genetic factor	-	[39]
Tuberculosis	4th	*Mycobacterium tuberculosis*, *Mycobacterium africanum*, *Mycobacterium bovis,* and *Mycobacterium microti*	Bacteria	[40]
Pneumonia	5th	Adenoviruses, Parainfluenza viruses, Influenza viruses, Measles virus, Herpes simplex virus, Respiratory syncytial virus, Coronavirus	Viruses	[32,41]
		*Cryptococcus neoformans*,	Bacteria	
		*Histoplasma capsulatum*, *Candida albicans*, *Aspergillus* spp.	Fungi	
Influenza	6th	Influenza A virus, Influenza B virus, Influenza C virus, Influenza D virus	Viruses	[42]
Lung cancer	7th	Human Papilloma virus, Epstein-Barr virus, BK virus, JC virus, Human Cytomegalovirus, Simian virus 40, and Measles virus, Human Herpesvirus 8, Human immunodeficiency virus,	Viruses	[43]
		*Chlamydia pneumonia*	Bacteria	
* COVID-19	8th	Severe Acute Respiratory Syndrome Coronavirus 2 (SARS-CoV-2)	Virus	[44]
Upper and lower respiratory tract infections (Pharyngitis, tonsillopharyngitis Epiglottitis and laryngotracheitis, Bronchitis and bronchiolitis)	9th	*Streptococcus pneumonia*, *Mycoplasma pneumonia*, *Haemophilus influenza* type b, *Corynebacterium diphtheriae*	Bacteria	[32]
		Parainfluenza virus, Epstein-Barr virus (EBV), Herpes Simplex virus, Coronavirus, Rhinovirus, Respiratory syncytial virus (RSV), Parainfluenza viruses, Adenoviruses, Herpes simplex virus	Viruses	
		*Candida albicans*	Fungi	
Others: Common cold	10th	Rhinoviruses, Parainfluenza viruses, Influenza viruses, Coronavirus, Respiratory syncytial virus	Viruses	[45,46]
Oral candidiasis		*Candida albicans*	Fungi	

* Ranking position subject to change due to the ongoing COVID-19 pandemic.

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
