# Peer review of "Plant-Derived Natural Products as Lead Agents against Common Respiratory Diseases"

_molecules, 2022, doi:10.3390/molecules27103054_

Round 1

Reviewer 1 Report

In this review, the author attempted to demonstrate the key role and prospect of natural products of plant origin in the discovery of new respiratory drug. Respiratory diseases (RDs) constitute a major public health problem worldwide especially the novel coronavirus (COVID-19) disease amongst others during the pandemic more recently. The topic is interesting and but the article has several considerable shortcomings, which prevents its publication in Molecules. What is more important in my opinion is to improve the quality of the paper which at present reads rather like a catalogue of management strategies such as symptomatic or specific therapy but not correlated with the current topic “natural products”. First of all, I wish to see the use of specific natural products from herbs and spices with potential significant activities as lead compounds for the treatment of RDs, i.e. the EC50 or SI data should compared with effects such as monoclonal antibodies on clinical usage. Likewise, authors analyzed mechanisms of action of natural lead agents against RDs, NF-κB and MAPK signaling pathways absolutely can not cover all the mechanisms targeting “the epithelial-mesenchymal transition (EMT), oxidative stress, fibroblast activation, inflammatory injury, metabolic regulation, and extracellular matrix accumulation within the respiratory tracts” etc. Also, there are many typographical and grammatical errors.

Author Response

REVIEWER 1 COMMENT

In this review, the author attempted to demonstrate the key role and prospect of natural products of plant origin in the discovery of new respiratory drug. Respiratory diseases (RDs) constitute a major public health problem worldwide especially the novel coronavirus (COVID-19) disease amongst others during the pandemic more recently. The topic is interesting and but the article has several considerable shortcomings, which prevents its publication in Molecules. What is more important in my opinion is to improve the quality of the paper which at present reads rather like a catalogue of management strategies such as symptomatic or specific therapy but not correlated with the current topic “natural products”. First of all, I wish to see the use of specific natural products from herbs and spices with potential significant activities as lead compounds for the treatment of RDs, i.e. the EC50 or SI data should compared with effects such as monoclonal antibodies on clinical usage. Likewise, authors analyzed mechanisms of action of natural lead agents against RDs, NF-κB and MAPK signaling pathways absolutely can not cover all the mechanisms targeting “the epithelial-mesenchymal transition (EMT), oxidative stress, fibroblast activation, inflammatory injury, metabolic regulation, and extracellular matrix accumulation within the respiratory tracts” etc. Also, there are many typographical and grammatical errors.

RESPONSE to Reviewer 1:

The manuscript has been improved by highlighting specific herbs and spices with potential significant activities as lead compounds for the treatment of respiratory diseases, mentioned in terms of their IC50, EC50 and MIC compared to their control drugs.

Specifically, some structure-activity relationship reports on natural anti-respiratory products have been added (lines 353-398), which includes table 4.

The mechanism of action section of the manuscript has been improved (lines 424-473), which includes table 5.

The manuscript has been cross-checked for typographical and grammatical errors.

Reviewer 2 Report

In my opinion, the presented work does not fullfil the journal requirements. There should be more emphasis on "molecules" rather than on diseases, their diagnosis and therapy. I am not saying that it is unnecessary, but the main emphasis should be on active structures and their mechanisms of action. Meanwhile, we have 1.5 pages in the manuscript on mortality and treatment costs that should be considered marginally. But the weakest chapter of the work is the mechanisms of action of potential therapeutics of natural origin. This fragment should be as developed as possible, with a detailed the mechanisms of action and the molecular target of the described natural compounds. I believe that both those compounds that are already used as therapeutics in the treatment of respiratory diseases, in clinical trials and in basic research (cell cultures). The authors have treated this issue marginally, and this should be the essence of the content of "Molecules". Only the presentation of the structures of relationships (of which there are in fact too many) does not contribute anything, because we do not see any structure-activity relationship. In my opinion, information on activities should be collected in an appropriate table.

Author Response

REVIEWER 2 COMMENTS:

In my opinion, the presented work does not fullfil the journal requirements. There should be more emphasis on "molecules" rather than on diseases, their diagnosis and therapy. I am not saying that it is unnecessary, but the main emphasis should be on active structures and their mechanisms of action. Meanwhile, we have 1.5 pages in the manuscript on mortality and treatment costs that should be considered marginally. But the weakest chapter of the work is the mechanisms of action of potential therapeutics of natural origin. This fragment should be as developed as possible, with a detailed the mechanisms of action and the molecular target of the described natural compounds. I believe that both those compounds that are already used as therapeutics in the treatment of respiratory diseases, in clinical trials and in basic research (cell cultures). The authors have treated this issue marginally, and this should be the essence of the content of "Molecules". Only the presentation of the structures of relationships (of which there are in fact too many) does not contribute anything, because we do not see any structure-activity relationship. In my opinion, information on activities should be collected in an appropriate table.

RESPONSES TO REVIEWER 2:

The “molecules” aspect of the manuscript has been improved (Lines 360-480). A new section has been added (lines 353-401), which includes table 4, highlighting the structure-activity relationships of natural products implicated in respiratory diseases.

The section on “mechanisms of action” of some potential natural products implicated in respiratory diseases, has also been expanded (lines 424-473). The mechanisms of action of some potential natural molecules against respiratory diseases are presented in table 5.

Thank you.

Reviewer 3 Report

I have marked some corrections/comments. The manuscript seems a good overview of the subject and is well organised and clear. Figure 7 chemical structures seem poor quality in my version. I would be inclined not to use plant latin name authorities (e.g. L.) in the text as they are in the tables/figures and many of your names in the text do not have them. I wonder if you can identify more clearly what results are in vivo and what are in vitro? 

Author Response

REVIEWER 3 - COMMENTS:

I have marked some corrections/comments. The manuscript seems a good overview of the subject and is well organised and clear. Figure 7 chemical structures seem poor quality in my version. I would be inclined not to use plant latin name authorities (e.g. L.) in the text as they are in the tables/figures and many of your names in the text do not have them. I wonder if you can identify more clearly what results are in vivo and what are in vitro? 

RESPONSES TO REVIEWER 3:

The marked corrections and comments have been attended to accordingly.

Figures 7a – 7d have been re-formatted for more clarity.

Latin name authorities have been expunged from the main text and left solely in the tables as advised.

The models of biological activities (in silico, in vitro, and in vivo) reported in this review, especially from section 5 (Structure-activity relationships of some implicated natural products (NPs) --- lines 353-401) to section 6 (Mechanisms of action of some implicated NPs – lines 424-473), have been stated.

Thank you

Reviewer 4 Report

The manuscript is a very detailed and precisely elaborated study in which authors compiled a considerable number literature sources on individual types of respiratory diseases, current treatment, management of diseases and subsequently provided a list and effects of natural substances with therapeutical potential against respiratory diseases. 

Author Response

REVIEWER 4 - COMMENT:

The manuscript is a very detailed and precisely elaborated study in which authors compiled a considerable number literature sources on individual types of respiratory diseases, current treatment, management of diseases and subsequently provided a list and effects of natural substances with therapeutical potential against respiratory diseases. 

RESPONSE TO REVIEWER 4:

No major issue was raised. But the manuscript has been revised accordingly.

Thank you.

Round 2

Reviewer 2 Report

The authors complied with the comments. I recommend the manuscript to appear in Molecules.

Author Response

Comments and Suggestions for Authors

The authors complied with the comments. I recommend the manuscript to appear in Molecules.

Response to the Reviewer

The manuscript has been checked again for editorial and grammatical errors and corrections have been made accordingly.

Thank you.